# High Spatial but Low Temporal Variability in Ectomycorrhizal Community Composition in *Abies alba* Forest Stands

**DOI:** 10.3390/microorganisms13020308

**Published:** 2025-01-30

**Authors:** Tina Unuk Nahberger, Hojka Kraigher, Tine Grebenc

**Affiliations:** Slovenian Forestry Institute, Večna pot 2, 1000 Ljubljana, Slovenia; hojka.kraigher@gozdis.si (H.K.); tine.grebenc@gozdis.si (T.G.)

**Keywords:** ectomycorrhizal community, spatiotemporal variation, silver fir, fungal diversity, forest, root-associated fungi

## Abstract

The ectomycorrhizal symbionts of silver fir have rarely been analyzed and identified, so little is known about their diversity and distribution. The aim of this study was (1) to analyze the diversity of ectomycorrhizal fungal species in three geographically distinct forest stands of *Abies alba* and (2) to demonstrate the high temporal variability of the ectomycorrhizal community over two consecutive growing seasons using repeated monthly sampling. Root samples were taken every month during two growing seasons in three silver fir-dominated forest stands. The ectomycorrhizal root tips were first assigned to a morphotype based on morphological characteristics and then identified by sequencing the internal transcribed spacer region. Alpha and beta diversity differed significantly between all three study sites, with the most diverse and even ectomycorrhizal community described in plot Jelovški boršt. The diversity indices over the growing season were different at two of the three study sites, supporting the idea of a fluctuation of ectomycorrhizal taxa during the growing seasons of the two consecutive years. While significant temporal variability was only confirmed for certain ectomycorrhizal taxa, there were no significant changes in the ectomycorrhizal community in general. Thus, we confirmed the high spatial but low temporal variability of the ectomycorrhizal community associated with silver fir.

## 1. Introduction

The silver fir tree (*Abies alba* Mill.) is considered one of the fundamental forest tree species for maintaining high biodiversity in forested ecosystems and one of the most important ecological and functional balancers of European forests [1]. Despite its importance in European forests, little is known still about its ectomycorrhizal (ECM) symbionts. Most of the conducted studies were performed in Poland [2,3,4,5,6], while studies focusing on the ECM symbionts of silver fir in other parts of Europe are scarce. Similar to Central Europe, silver fir has been exposed to a considerable decline in the Balkan peninsula as well [7,8,9], due to different factors, e.g., forest management, animal browsing, and, more recently, climate warming [1,7]. Therefore, knowledge about the range of silver fir ECM symbionts in different areas is even more important for us to understand how the ECM symbionts of silver fir adapt to different conditions. The spatial distribution of individual ECM fungal taxa is driven by yet not well correlated with environmental factors such as soil conditions, local climatic characteristics, and geographical location [10,11]. When changes in space usually lead to dramatic changes in several biotic and abiotic factors, seasonal changes usually cause alternating climatic conditions by affecting nutrient transfers as a result of changes in plant physiology [12]. It has been well established that fungal diversity and community composition can vary across temporal scales such as days [13], months [14], seasons [15,16,17,18], and years [19]. Variations in fungal biomass [16], community composition, and species richness in accordance with temporal/seasonal variations have already been reported [14,16,17,19,20,21].

The temporal variations of ECM fungi are most likely tightly linked to the environmental conditions and the phenology of trees, with light, soil pH, soil nutrients, temperature, and moisture as the main abiotic drivers [22]. The between season variations in environmental conditions and the corresponding ECM fungal taxa preferences for different conditions may favor the growth of certain ECM fungal taxa over others, resulting in intra-annual shifts in ECM communities [23,24,25,26]. In addition, the seasonal fluctuation of fine root growth has been implicated as a potential driver of the seasonal patterns in ECM communities [27,28]. The ecosystem significance of seasonal changes in soil mycorrhizal community composition depends partly on whether these changes affect soil nutrients or carbon cycling [29].

Based on the literature data, the seasonal or temporal variations in ECM fungi associated with silver fir have, unfortunately, not been analyzed yet. Most of the studies focused on beech, oak, pine, and a minor portion focused on some other tree species, e.g., lindens, birch, hornbeam, etc. To understand the plasticity of the spatial–temporal variations of the ECM communities associated with silver fir, we used a DNA-based approach to analyze the spatial variations in the ECM community associated with silver fir at three geographically different locations in Slovenia, where the temporal variations of the alpha and beta diversity of the ECM community were followed on a monthly basis for two consecutive growing seasons.

## 2. Materials and Methods

### 2.1. Field Plots

Three different plots were located within the central natural distribution of the silver fir trees in Slovenia, namely Jelovški boršt (JB), Lehen na Pohorju (LP), and Ljubelj (L) (Table 1). At each location, a representative plot (50 m × 50 m) with silver fir as a dominating tree was established. In each plot, five healthy and undamaged adult silver fir trees were selected. The trees were at least 80 years old, with a minimum diameter at breast height of 31 cm. The selected trees were at least 7 m apart to minimize sampling in the same common mycorrhizal network (sensu [30,31,32,33]).

### 2.2. Sample Collection

Root samples were collected from the soil cores (10 cm in diameter) of the 0–20 cm deep soil layer close to the tree trunk (at distances ≤ 1 m), by using a soil core for each tree. This study was conducted during two consecutive growing seasons from March–October (also in November at JB), in the years 2016 and 2017. For comparable data, sampling at all three sites (JB, L and LP) was performed within two days in each sampling month. Altogether, 5 soil samples per location × month l were taken, which resulted in 15 soil samples per month and a total of 240 soil samples. Individual samples were placed in a plastic bag and stored at 4 °C before being processed.

### 2.3. ECM Morphological Analyses

Samples were soaked in tap water for approximately 15–20 min to separate the roots from the soil. All roots were further gently washed and divided into coarse and fine roots, as defined by Železnik et al. [32,34], who considered fine roots as roots with a diameter less than 2 mm. Fine roots were transferred into a Petri dish filled with tap water and were randomly picked and investigated under an Olympus SZX12 stereomicroscope (Olympus, Hamburg, Germany). In each sample, 500 root tips were investigated (including non-vital ECM or non-ECM root tips) and counted. Dead root tips were distinguished from vital roots tips by their turgidity and color. The vitality of ECM root tips was expressed as the ratio of vital root tips to total number of analyzed root tips. The ECM root tips were assigned to a morphotype based on morphological features (color and surface structure of the mantle, ramification type, presence, and abundance of the emanating element) according to the methodology of Agerer [35]. All morphotypes were photographed under a stereomicroscope. The most common ECM morphotypes were described in our previous study, Unuk Nahberger et al. [36]. Three to five root tips per morphotype were freeze dried and further processed for DNA analysis to obtain molecular identities.

### 2.4. DNA Extraction and Molecular Identification of ECM Fungal OTUs

ECM fungal OTUs were identified by sequencing the internal transcribed spacer (ITS) region of each morphotype. Total DNA was extracted using the DNeasy Plant Mini kit (Qiagen, Hilden, Germany) following manufacturer’s instructions. The ITS region of nuclear ribosomal DNA was amplified from isolated DNA using the fungus specific primer pair ITS1F and ITS4 [37,38], following the modified procedure described in [39]. Sangar sequencing was performed by a commercial sequencing laboratory (Macrogen Inc., Seoul, Republic of Korea). The acquired sequences underwent alignment and analysis using bioinformatic software Geneious (v.11.1.4; [40]). Following this, a comparison with other reference sequences via a BLASTN algorithm from the National Center for Biotechnology Information website (https://blast.ncbi.nlm.nih.gov/Blast.cgi, accessed on 6 December 2018) and from the UNITE website (https://unite.ut.ee/analysis.php, accessed on 6 December 2018) was conducted. Sequences that remained unclassified at a family or higher taxonomic level were discarded. Final criteria for a database match were as followed: query cover ≥ 80% and sequence similarity > 92% (representing an approximate cut-off value at genus level) or sequence similarity ≥ 97–100% (representing an approximate cut-off at species level) [41,42].

ECM status of obtained sequences and corresponding taxonomic units was determined according to Rinaldi et al. [43] and Tedersoo et al. [44]. Sequences of ECM fungi were deposited in the GenBank database under accession numbers MN265475–MN265819.

The relative abundance of each morphotype (RA_x_) was calculated using the following formula:$$RAx=\frac{number\;of\;root\;tips\;with\;morphotype\;x}{number\;of\;total\;root\;tips\;examined}$$

### 2.5. Statistical Analyses

All statistical analyses were performed in R version 3.5.1 (R Core Team, 2016 [45]). Data were visualized with RStudio using the ggplot2-package [46]. Alpha diversity indices (assessed by Hill’s numbers; Hill, 1973 [47]) were calculated using the vegan package [48]. We used the most commonly used diversity indices: OTU richness, inverse Simpson’s index, which calculates the proportional abundance of each species and is interpreted as equivalent to OTU evenness, and the inverse Berger–Perker index, which calculates the proportional abundance of the most abundant species and can be interpreted as OTU dominance. Nested ANOVA with a sampling month nested in a sampling year was used to observe the differences in alpha diversity and vitality of ECM root tips throughout the sampling time. The homogeneity of variance assumptions was checked with a Levene test. Venn analysis, with the function trans_venn from microeco package [49] was used to obtain shared and unique taxa across the sampling sites.

Effects of sampling location, year, and month on ECM fungal community composition were tested using adonis function on Bray–Curtis matrix and 999 iterations, in the vegan package [48].

Multivariate generalized linear models (MV-GLMs) on Poisson regression, specifically, the mvabund and manyglm functions from the mvabund package [50] were used to examine differences in ECM fungal OTU abundances throughout the sampling time. Multivariate and unadjusted univariate *p*-values were obtained by Wald tests, both using 10,000 Monte Carlo permutations.

## 3. Results

Within the 240 *Abies alba* root samples pooled together, we were able to identify 92 different ectomycorrhizal OTUs, of which, 13.1% belong to Ascomycota and 86.9% to Basidiomycota.

The Venn diagram (Figure 1) and PERMANOVA showed that the ECM fungal community was highly diverse among the study sites (*p* < 0.05). Fifty-nine OTUs were recorded at study site Jelovški boršt, 27 of them were shared with study site Ljubelj and 20 with study site Lehen na Pohorju. Study site Lehen na Pohorju hosted 41 ectomycorrhizal OTUs, 25 of them were shared with study site Ljubelj, which hosted 47 ectomycorrhizal OTUs. Only 17 ectomycorrhizal OTUs were shared between all three study sites (Figure 1). In order of abundance, the most abundant ECM fungal OTUs were assigned to *Russula*, *Cenococcum*, *Tomentella*, and *Lactarius*, with the average abundance higher than 10% per sample.

At Jelovški Boršt (JB), the following common ECM fungal OTUs with high relative abundance (RA) were identified: *Cenococcum geophilum* (RAmean = 11.65%), *Lactarius salmonicolor* (RAmean = 8.11%), *Tomentella stuposa* (RAmean = 5.20%), *Sebacina epigaea* (RAmean = 2.78), and *Russula illota* (RAmean = 2.58%), respectively (Figure 2).

The most abundant ECM fungal OTUs at study site Ljubelj (L) were *Cenococcum geophilum* (RAmean = 7.24%), followed by *Russula chloroides* (RAmean = 5.64%), *Amanita rubescens* (RAmena = 4.43%), *Tricholoma virgatum* (RAmean = 4.42%), and *Clavulina coralloides* (RAmean = 4.1%) (Figure 2).

In order of abundance, study site Lehen na Pohorju (LP) hosted the following abundant ECM OTUs: *Thelephora wakefieldiae* (RAmean = 6.41%), *Russula ochroleuca* (RAmean = 4.95%), *Elaphomyces granulatus* (RAmean = 4.39%), and *Xerocomellus pruinatus* (RAmean = 4.29%) (Figure 2).

The sampling year also represented a significant factor in shaping the ECM fungal community structure (PERMANOVA, *p* < 0.05). Sixty-nine OTUs were recorded in 2016 and sixty-three in 2017, wherein forty of them were shared.

### Temporal Dynamic of Ectomycorrhizal Community

The proportion of vital fine root tips colonized by ECM fungi remained relatively stable through time, although some spring–summer–autumn patterns were revealed (Figure 3).

The sampling year did have a significant effect on alpha diversity (all three analyzed diversity) indices at sampling sites Jelovški boršt and Lehen na Pohorju (*p* < 0.05) also, but it did not impact alpha diversity at Ljubelj (Table 2). Further, the sampling month significantly influenced only species richness at site Jelovški boršt, but not the evenness and dominance. In contrast, at site Lehen na Pohorju, significant differences in all three diversity indices throughout the sampling months were confirmed (*p* < 0.05). Similar to sampling year, no significant differences were observed at site Ljubelj throughout the sampling months. Diversity indices (mean ± stderr.) of the temporal dynamics for the ECM of *Abies alba* in the samples from the study sites Jelovški boršt, Ljubelj, and Lehen na Pohorju are presented in Appendix A. Similar to the dynamics of ECM root tips, some spring–summer–autumn patterns were observed for richness as well (Figure 4).

In general, there were no significant changes in ECM community structure throughout the sampling months (PERMANOVA, *p* > 0.05 for all three study sites); despite this, we did confirm significant changes in the abundances of several abundant OTUs at the individual sampling sites, throughout the sampling time (Figure 5, Appendix A). At sampling site Jelovški boršt, significant differences in the abundances of several *Russula* species were observed, e.g., *R. cyanoxantha* (was significantly the highest in March 2016 and 2017), where for *R. illota*, the highest abundances were observed in the spring and autumn of 2017, with low abundances in 2016 compared to *R. nigricans*, which had the highest abundances during the spring and autumn of 2016, with low abundances in 2017. Interestingly and contradictory to the Jelovški bortš site, the higher abundances of *R. nigricans* at the Ljubelj site were observed in the spring and autumn of 2017. The abundance of *A. rubescens* at Ljubelj indicates a more significant occurrence in late summer and autumn, whereas a similar occurrence pattern was also confirmed for *R. ochroleuca*. For *A. byssoides*, the significant impact of sample month was confirmed, as a tendency to occur in higher abundances during the spring (e.g., May) and autumn (e.g., September/October) was observed at the Jelovški boršt and Ljubelj sites. Similarly, *S. epigaea* occurred in higher abundances during the spring in both the analyzed years at the Jelovški boršt and Ljubelj sites, when surprisingly at Lehen na Pohorju higher abundances during the summer of 2016, with very low abundances in the year 2017, were recorded. As for site Lehen na Pohorju, completely different occurrence patterns and ECM community structures were observed between the sampling years 2016 and 2017. Several OTUs were observed in higher abundances in the sampling year 2016, when their abundances were very low in 2017, e.g., *L. salmonicolor*, *T. stuposa*, *T. fibrillose*, and *X. pruinatus*, compared to the OTUs *E. granulatus*, *R. ochroleuca*, and *T. wakefieldiae*. Despite that, several OTUs at the Lehen na Pohorju site did show a similar occurrence tendency for both years, e.g., *B. erythropus* (spring and autumn), *L. subdulcis* (late spring–autumn), and *T. virgatum* (spring), which a similar occurrence pattern was observed as well at the Ljubelj site.

## 4. Discussion

Our study reports two key results: first, a highly diverse ECM fungal community among the different study sites and second, the temporal variation of ECM fungi together with alpha diversity. By pooling the sequence data of 240 *Abies alba* root samples, we were able to identify 92 different ECM root-associated OTUs. As was already reported by our previous study, Unuk and Grebenc [51], and by recently published studies [5,52], the observed results confirm the high potential for diversity of ECM fungi in silver fir stands, as the number of identified ECM symbionts of silver fir increases with the newest studies, which may be partly attributed to the accessibility and development of DNA sequence-based methods in recent decade. Our results also show the high spatial specificity of the ECM symbionts of silver fir, as out of the 92 identified ECM OTUs, only 17 ECM taxa occurred at all three sampled plots; meanwhile, 29 (JB), 13 (LP), and 12 (L) ECM taxa were specific to each individually sampled plot (Figure 1). Taken together, in our study, the species richness of ECM taxa decreased with elevation, as the highest number of ECM taxa was identified at site JB (59 ECM taxa), which was the lowest sampled site (Table 1). Despite the fact that several researchers confirmed a decline in fungal species richness towards the poles or high latitudes [12,53,54], in accordance with the general spatial distribution pattern, several studies showed on a lack of a consistent spatial distribution pattern for fungal diversity [55,56,57], which was also confirmed by our results as ECM species richness did not decline at a higher latitude. The observed differences in ECM assemblages between the sampled sites are probably due to differences in edaphic factors and environmental conditions (e.g., air temperature and precipitation), which was already reported by several authors [12,17,18,58]. As sites JB and L are characterized by similar edaphic factors, site LP stood out because of the higher organic C and total N contents, thereby we suggest that the observed differences in ECM community compositions are driven by edaphic factors rather than latitude. Despite minor differences, the JB and L sites showed similar ECM community compositional structures, which cannot be said for the site LP, where among others the significantly higher abundances of members from the family Boletaceae were recorded. The specific occurrence of OTUs from the Boletaceae family may be linked to the higher C:N ratio at site LP, which is in accordance with the results of Martinez-Peña et al. [59], as the authors in this study confirmed the positive correlation of ECM taxa from the genus *Boletus* with sandy soil and the C/N ratio.

Although we expected to find similar ECM fungal occurrence patterns and community compositions based on the sampling time, our results indicated that ECM richness and communities are not seasonally static across years, as was already reported by Mundra et al. (2015). The comparison of alpha diversity between two consecutive sampling years showed significant changes in alpha diversity and the ECM root tips number. A similar result was observed for ECM community composition, as the ECM assemblages of fungal OTUs, recorded in the first sampling year, differed from those obtained in subsequent year. Despite the differences between sampling years, the ECM root tip number and alpha diversity remained relatively stable throughout the individual year; however, we did observe some spring–summer–autumn patterns, with a higher ECM root tip number and higher ECM species richness during the spring and autumn, with a decline in the summer. The observed patterns may be a result of the seasonal variation of root elongation and fine root growth, with major periods of growth during the budburst in the spring and after leaf senescence in the autumn [14,23,60,61] and fine root inactivity and decay in the summer [62,63,64]. In contrast to our results, several authors observed higher fungal diversity during the summer (e.g., July, August) when the trees possessed the most vigorous growth and thereby potentially provided more C assimilate to the fungi [16,17]. It has been suggested that ECM fungal diversity seems to be tightly associated with, on the one hand, seasonal changes in photosynthetic activity in accordance with host phenology, (throughout the seasonal variation of belowground carbon allocation to the tree roots [16,17,21,65,66], and, on the other hand, with the different weather conditions within seasons.

The beta diversity of the total ECM fungal community was unaffected by the sampling time, e.g., sampling month; however, we did observe changes in relative abundances in several individual ECM OTUs. In contrast to Voriškova et al. [16], most of the ECM OTUs, for which temporal variations of relative abundances were confirmed, showed higher relative abundances either during the spring and/or the autumn months, whereas Voriškova et al. [16] observed higher abundances of most of the ECM taxa during the summer. In our study, the OTUs assigned to several *Russula* species (e.g., *R. cyanoxantha*, *R. illota*, *R. nigricans*) had greater abundances during spring and autumn; meanwhile, *R. ochroleuca* had greater abundances in late summer and autumn. Richard et. al. [67] observed higher abundances of the *Russula* species during autumn, but not also during the spring months, whereas, in the study by Voriškova et al. [16], lower abundances of the *Russula* species were observed during spring. Similar to Voriškova et al. [16], higher abundance of the OTU assigned to *A. rubescens* were observed in the late summer and autumn. For *S. epigeae*, different occurrence patterns were observed, e.g., a higher abundance in April at the Jelovški boršt and Ljubelj sites, versus a higher abundance in July at Lehen na Pohorju, which was in accordance with the findings of Gorfer et al. [18], with higher abundances of Sebacinales observed in August. In accordance with our results, typical occurrence patterns of several ECM fungi, e.g., *L. subdulcis* (summer, autumn), *T. stuposa* (throughout the sampling time), and *X. pruinatus* (spring and autumn) were also observed in the study of Khokon et al. [68]. As was already speculated, the ECM taxa that occur with greater abundance in the spring could rely less on recently assimilated C and potentially be replaced by other ECM taxa later in the growing season [18]. During autumn and early winter, increased soil organic matter [17,69,70,71] could support the growth of ECM fungi with saprotrophic ability in their roots [69]; however, there is still a lack of the evidence to confirm this function of ECM fungi.

The observed temporal differences in the relative abundances of several ECM taxa are likely partly caused by the seasonal variation in soil temperature and water content. Changes in temperature, moisture, and litter decomposition have been recognized as the key drivers shaping temporal ECM fungal assemblages [16,17,55,72,73], via directly controlling the enzymatic production and activities of fungi [74] and indirectly influencing the plant community and soil nutrients [73,75,76]. Due to a lack of variability in the soil chemistry and climatic data throughout the sampling time, it is unclear whether the observed patterns in our study could be driven by a variation in specific edaphic and/or climatic conditions.

As most of the conducted studies analyzed the ECM community structures between seasons, but not throughout the months, our study demonstrated that shifts in ECM community composition can change from one month to another, suggesting the great importance of sampling time in analyzing ECM fungal community composition. Several authors already suggested the importance of repeated samplings of fungal species to increase the chances of detecting species that previously remained undetected due to seasonal variations within fungal communities, among others [77,78]. The latter was also confirmed in our study, as several of the more dominant ECM species occurred only at certain times, e.g., a specific month, but were undetectable in other months.

## 5. Conclusions

In conclusion, this study indicates the high spatial but low temporal variability of ECM communities in three geographically distinct *Abies alba* forest stands in Slovenia. Significant differences in ECM community structures between the studied plots were confirmed, with only 17 ECM OTUs shared between all three study sites, indicating the high spatial variability of ECM communities. Despite the low temporal variability of the ECM community observed in this study, significant changes in the abundances of several abundant ECM OTUs at the individual sampling sites, throughout the sampling time, were observed. With the results obtained, we have shown that ECM communities can change very rapidly, and that repeated sampling is crucial when analyzing the ECM community structure associated with forest tree species.

## Figures and Tables

**Figure 1 microorganisms-13-00308-f001:**
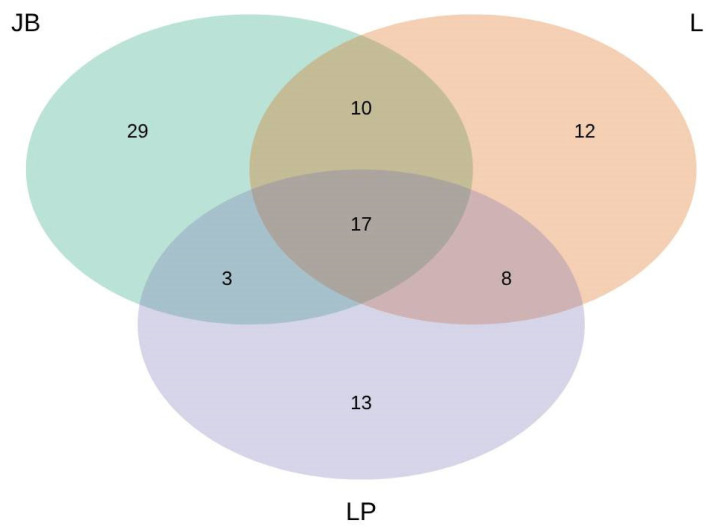
Venn diagram with number of shared and location-specific ECM OTUs per study site Jelovški boršt (JB), Lehen na Pohorju (LP), and Ljubelj (L).

**Figure 2 microorganisms-13-00308-f002:**
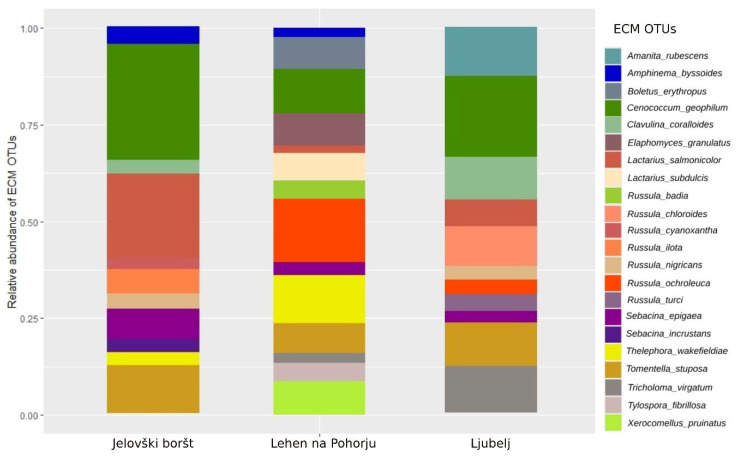
ECM community structure at analyzed sampling plots, e.g., Jelovški boršt (JB), Lehen na Pohorju (LP), and Ljubelj (L). Only ECM fungal OTUs that occurred at least 10 times and in at least 5% of all soil samples are presented.

**Figure 3 microorganisms-13-00308-f003:**
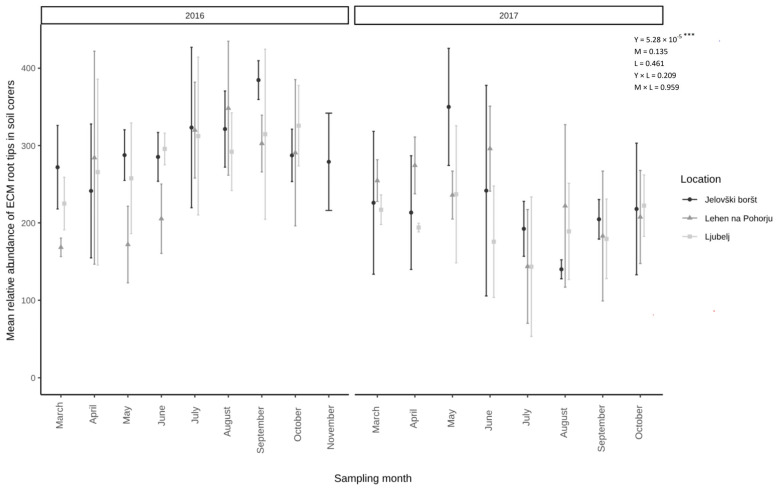
Temporal dynamic of vital ECM root tips in soil cores from study sites Jelovški boršt, Ljubelj, and Lehen na Pohorju in two consecutive years. Nested ANOVA results (*p*-values) for L—location (JB-L-LP), Y—year (2016 vs. 2017), M—sampling months (April till October/November), and Y × L and M × L are presented. An asterisk (***) indicated a significant result (*p* < 0.0001).

**Figure 4 microorganisms-13-00308-f004:**
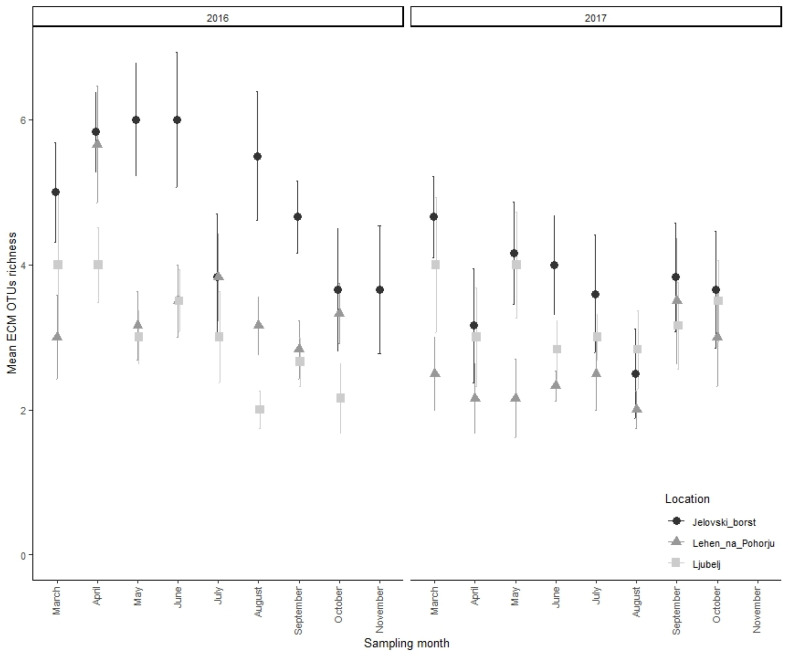
Temporal dynamics of ECM OTUs richness at study sites, Jelovški boršt, Lehen na Pohorju, and Ljubelj. The mean ECM fungal OTU richness ± standard deviation per sampling month is shown.

**Figure 5 microorganisms-13-00308-f005:**
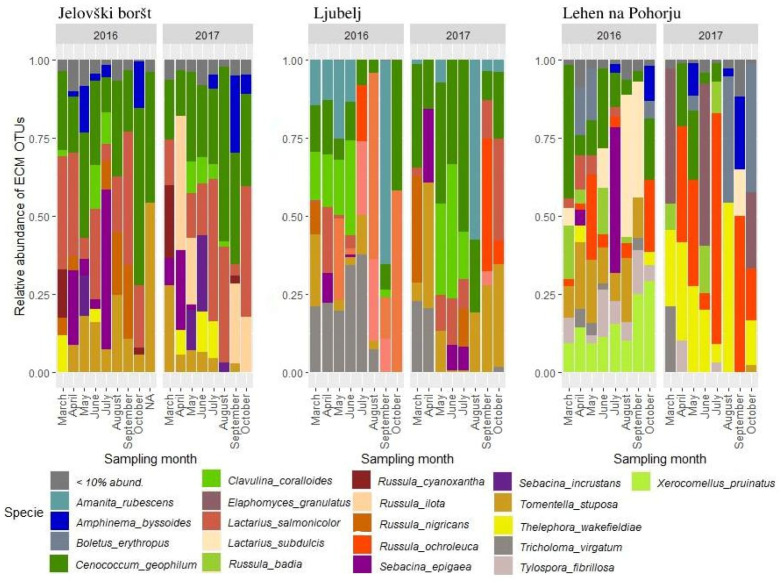
ECM community composition shifts at a monthly time scale at three sites. Only ECM fungal taxa that occurred at least 10 times in the years of sampling and in at least 5% of all soil samples are shown. Rare ECM fungal taxa with less than 10% in a given sampling time were pooled in a single group named <10% abund. The significance test data for each individual ECM fungal taxon are given in Appendix A.

**Table 1 microorganisms-13-00308-t001:** Characteristics of all three analyzed study sites in Slovenia.

Study Site	Coordinate (Gauss)	Climate	Altitude (m a.s.l)	Forestry Classification	pH 0.01 M (CaCl_2_)	Organic C (%)	Total N (%)
Jelovški boršt (JB)	45.45° N, 15.05° E	Temperate continental climate	180–225	Dryopterido-Abietetum	4.04	2.83	0.17
Ljubelj (L)	46.24° N, 14.15° E	Alpine climate	960–1050	Dryopterido-Abietetum	3.5	1.40	0.09
Lehen na Pohorju (LP)	46.33° N, 15.20° E	Alpine climate	469–611	Dryopterido-Abietetum	3.5	10.30	0.62

**Table 2 microorganisms-13-00308-t002:** Diversity indices nested ANOVA results (*p*-values) for sampling month (April till October/November), sampling year (2016 vs. 2017), and sampling month × sampling year are presented. An asterisk (*, ** and ***) indicates a significant result.

		Richness	Evenness	Dominance
	Sampling month	0.00113 **	0.143	0.302
Jelovški boršt	Sampling month × factor (Sampling year)	0.00173 **	0.00299 **	0.00545 **
	Sampling month	0.00181 **	0.00181 **	0.0102 *
Lehen na Pohorju	Sampling month × factor (Sampling year)	2 × 10^−4^ ***	0.00172 **	0.015 *
	Sampling month	0.319	0.202	0.194
Ljubelj	Sampling month × factor (Sampling year)	0.392	0.239	0.298

## Data Availability

The original contributions presented in this study are included in the article/Appendix A. Further inquiries can be directed to the corresponding author.

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
