# Peer review of "High Spatial but Low Temporal Variability in Ectomycorrhizal Community Composition in Abies alba Forest Stands"

_microorganisms, 2025, doi:10.3390/microorganisms13020308_

Round 1
Reviewer 1 Report
Comments and Suggestions for Authors
Dear authors,
I read with great interest the research received for initial review „High Spatial But Low Temporal Variability in Ectomycorrhizal Community Composition in Abies alba Forest Stands”, this research addresses a relatively underexplored topic: the diversity and distribution of ectomycorrhizal symbionts in silver fir (Abies alba). Given the limited studies on this subject, the research offers significant novelty. It analyzes ectomycorrhizal diversity across three geographically distinct forest stands, with a focus on temporal variability over two consecutive growing seasons. The use of morphological characterization and ITS sequencing provides detailed insights into the community structure. The study reveals high spatial diversity but lower temporal variability in ectomycorrhizal communities, highlighting the influence of local factors. These findings have scientific potential, especially in forest management and future symbiotic interaction studies.
However, I have substantial concerns regarding the experimental design in the context of this research, particularly the fact that the root samples were collected between 2016 and 2017. Given the nearly nine years that have passed since the collection of the samples, is it plausible to assume that the analyzed results have not undergone significant changes over time? Considering this extended period, can the research still be regarded as novel? Furthermore, is there any existing research that supports the notion that the results retain their novelty after such a long duration, without being significantly affected by the passage of time?
I appreciate the authors in advance for any responses they may provide.

Author Response
Dear Reviewer,
First of all, thank you very much for your interest in our study. I agree that there has been a long gap between the sampling and the writing of this manuscript, and I wish I could publish it sooner. However, since the root samples were obtained during the sampling for my PhD thesis, I had to focus on them first in order to complete my thesis. After completing my PhD in 2020, I was absent for a very long time due to two maternity leaves. As I had some health issues during both pregnancies, I could not complete this manuscript earlier, due to certain restrictions. After a long absence, I returned to work in mid-September 2024, focusing firstly on publishing these results, as I believe that this topic is still novel, since only a few articles on ECM communities of Abies alba have been published since 2017, none of which I have found focusing on the temporal variability of ECM communities associated with Abies alba. Thereby I believe presented data, still preserves the novelty and importance, even though they were obtained several years ago. Moreover, one of the main objectives of this study was to observe the processes in natural forest ecosystems, e.g. the interactions between symbiotic fungi and forest tree species, and their changes in short time intervals (e.g. months) and space, which are long-lived processes (decades to centuries) rather than the structure of ECM communities, which can actually change relatively quickly, as also presented in this study.
I strongly believe that the results are worth publishing, because as you mentioned our findings have some potential in future symbiotic interactions studies, not only in Abies alba forest stands, but also more broadly.
Reviewer 2 Report
Comments and Suggestions for Authors
Please follow the guidelines for authors, e.g. citations within the text of the manuscript should be using numbers not names of the authors, formatting of the references is not according the guidelines for authors, template is not exactly as the document of the manuscript.
Table 1: Please use dot not comma as a decimal separator of the numbers.
It is not clear how many replications were in each location in one date. The total number of samples (240) is not divisible by the number of measurement dates (17).
Please be more specific in description of the results presented in the figures and tables. For example, it is not clear what is presented in the legend of the Fig. 3. The values are P-values? Interactions should be presented rather as “YxM”, not “Y:M” (it is used only in R code, not in presentation of the results). Similar comment applies for Table 2.
It would be good if section Conclusion was added to the manuscript.
Author Response
Dear Reviewer,
thank you very much for taking your time to review our manuscript. Please find the responses below and the corresponding corrections in track changes in re-submitted files.
Comment 1: Please follow the guidelines for authors, e.g. citations within the text of the manuscript should be using numbers not names of the authors, formatting of the references is not according the guidelines for authors, template is not exactly as the document of the manuscript.
Response 1: Thank you for pointing out, we have now followed the journal guidelines for authors.
Comment 2: Table 1: Please use dot not comma as a decimal separator of the numbers.
Response 2: Was corrected, dots are used.
Comment 3: It is not clear how many replications were in each location in one date. The total number of samples (240) is not divisible by the number of measurement dates (17).
Response 3: We added the sentence about the number of replicant per each location in Materials and method, section Sample collection. Per each location, 5 soil samples per month were collected.
Comment 4: Please be more specific in description of the results presented in the figures and tables. For example, it is not clear what is presented in the legend of the Fig. 3. The values are P-values? Interactions should be presented rather as “YxM”, not “Y:M” (it is used only in R code, not in presentation of the results). Similar comment applies for Table 2.
Response 4: We corrected the description and the Figure 3. Presented are P-values, from nested ANOVA. Also in Table 2 we have corrected the interactions and added an explanation of presented P-values.
Comment 5: It would be good if section Conclusion was added to the manuscript.
Response 5: A Conclusion section was added.
Round 2
Reviewer 1 Report
Comments and Suggestions for Authors
Dear Authors,
I truly appreciate the honesty and effort reflected in your response. That being said, in the Results and Discussion section, I would kindly suggest better emphasizing how your results, obtained during the 2016–2017 period, are primarily comparable to findings already published from the specific areas of Jelovški boršt (JB), Ljubelj (L), or Lehenna Pohorju (LP). If that is not possible, perhaps drawing comparisons with results from nearby regions or studies published at the institutional level would be helpful. It is particularly important to align your findings with studies published in the last two to three years.
Additionally, based on your results and conclusions, I would like to ask whether it is positive—or potentially concerning—that, as you state, these results do not appear to have been influenced by the passage of time. I believe it would be valuable to propose a hypothesis based solely on your findings to explore this aspect further.
Thank you in advance for your efforts in revising this manuscript. However, we must ensure that the results obtained reflect the current reality.

Author Response
Dear Reviewer,
Ectomycorrhizal fungi associated with Abies alba are very much understudied, not only in Slovenia but also in Europe in general. At our institution I was the first one, focusing on Abies alba ectomycorrhizas and thereby studied location e.g. Ljubelj, Lehen na Pohorju and Jelovški boršt have not been studied nor before neither after my performed studies. This is why I believe that obtained results are so much more important, as were performed at locations and forest tree species that has not been studied before or are understudied, especially in Slovenia. The samplings were also performed in very short time intervals, which is very rare, as most of the published studies in which authors analyzed temporal variations of ectomycorrhizal fungi, performed only several samplings, most likely once per season. Our study stood out as the samplings were performed once per month for two consecutive growing seasons, and studies like this are very scarce.
If there would be any recently published studies on spatial-temporal variation of ectomycorrhizal fungi associated with Abies alba, I would align it with my findings, as is also in my best interest to compare our obtained results with those already published. Today I went through Google Scholar again, to check if there are any new published studies after 2020, focusing on spatial and temporal variation of Abies alba ectomycorrhizas. Unfortunately, I didn’t find them. Abes alba ectomycorrhizas have been analyzed mostly in Poland and Ukraine, which are due to completely different soil characteristics hard to compare with ours. The closest region in which sampling of Abies alba roots were performed is in Bosnia and Herzegovina, a study performed by my colleague and was published in 2020, however it was only a one-time sampling and I do not believe it is correct to align those results with ours, as the main purpose of our study was to analyze spatial-temporal variation of ectomycorrhizal community rather than the structure itself.
I went throughout several published studies of ectomycorrhizal fungi in different forest sites, to check the time gap between the sample collection and publishing, and I found quite a few studies, published in different journals, with similar time gap between sample collection and publishing the results. Here are just few of them: e.g. Mandolini et al. 2024, sample collection in 2008 (https://link.springer.com/content/pdf/10.1007/s11104-022-05497-z.pdf), Beidler et al. 2023, sample collection in 2016 (https://besjournals.onlinelibrary.wiley.com/doi/pdf/10.1111/1365-2745.14112), Ding et al. 2023, sample collection in 2017-2018 (DOI: 10.1111/1365-2745.14151), Bashian-Victoroff et al. 2025, sample collection in 2018 (https://doi.org/10.1016/j.funeco.2024.101388), Prieto-Rubio et al. 2024, sample collection in 2016 and 2017 (https://doi.org/10.5194/soil-10-425-2024) . However, in most of published studies authors do not even specify the sampling time, e.g. year, thereby for those published results we can not assume whether they’ve been significantly affected by the passage of time.
Again, I believe that our results have not been significantly affected by the passage of time, as samples were processed as short as one month after its collection (e.g. ECM morphotyping and sequencing). As the main purpose of this study was to demonstrate spatial-temporal variation of ectomycorrhizal fungi associated with host roots - a long-lived process in forest ecosystem, and for which I strongly believe are not influenced in time-period of less than 10 years. And as our results coincide well with already published ones (even with those from studies published after 2020), e.g. greater species diversity and higher abundances of most dominant ECM OTUs during spring and summer, and occurrence of individual ECM fungi taxa during specific time, e.g. month, mostly coincide well with published studies. Thereby I don’t think that our results were significantly affected by the passage of time and can reflect current situation in spatial-temporal variation of ECM fungi.
However, we have reformulated our hypothesis at the end of introduction section, so it focused only on spatial-temporal variation of ECM community associated with silver fir and not to describe the community structure itself.
I hope that with the written and with presented examples, I have satisfied your doubts and showed that our results have not been significantly influenced by the passage of time and can reflect current situation.
Thank you for your effort for review and comments to improve our manuscript.